# Metabolically Healthy Obesity: Presence of Arterial Stiffness in the Prepubescent Population

**DOI:** 10.3390/ijerph17196995

**Published:** 2020-09-24

**Authors:** Maria Isabel Ruiz-Moreno, Alberto Vilches-Perez, Cristina Gallardo-Escribano, Antonio Vargas-Candela, Maria Dolores Lopez-Carmona, Luis Miguel Pérez-Belmonte, Alejandro Ruiz-Moreno, Ricardo Gomez-Huelgas, Maria Rosa Bernal-Lopez

**Affiliations:** 1Internal Medicine Department, Instituto de Investigacion Biomedica de Malaga (IBIMA), Regional University Hospital of Malaga, 29010 Malaga, Spain; mruiz.salud@gmail.com (M.I.R.-M.); antonio.vargascandela@gmail.com (A.V.-C.); mdlcorreo@gmail.com (M.D.L.-C.); luismiguelpb@hotmail.com (L.M.P.-B.); alejandrorm0540@gmail.com (A.R.-M.); rgh@uma.es (R.G.-H.); 2Endocrinology and Nutrition Department, Instituto de Investigacion Biomedica de Malaga (IBIMA), University Hospital Virgen de la Victoria, 29010 Malaga, Spain; Vp88alberto@gmail.com; 3Clinical Analysis Department, Regional University Hospital of Malaga, 29010 Malaga, Spain; carmencge@gmail.com; 4Medicine and Dermatology Department, University of Malaga, 29010 Malaga, Spain; 5CIBER Fisiopatología de la Obesidad y la Nutrición, Instituto de Salud Carlos III, 28029 Madrid, Spain

**Keywords:** arterial stiffness, carotid-femoral pulse wave velocity, prepubescent population, Mediterranean diet, insulin resistance, metabolically healthy obesity

## Abstract

Aim: Arteriosclerotic cardiovascular disease, one of the world’s leading causes of death, first manifests itself at an early age. The identification of children who may have increased cardiovascular risk in the future could be an important prevention strategy. Our aim was to assess the clinical, analytical, and dietary variables associated with arterial stiffness (AS), measured by carotid-femoral pulse wave velocity (cfPWV) in a prepubescent population with metabolically healthy obesity (MHO). Subjects and Methods: A cross-sectional study in prepubescent subjects with obesity who had ≤1 metabolic syndrome criteria (abdominal perimeter and blood pressure ≥90th percentile, triglycerides >150 mg/dL, HDL-cholesterol <40 mg/dL, fasting plasma glucose ≥100 mg/dL) was conducted. Adherence to Mediterranean Diet, blood pressure, BMI, waist/height ratio (WHtR), glycemic status, lipid profile, and cfPWV were analyzed. 75 MHO children (boys: 43; girls: 32; *p* = 0.20) (age = 10.05 ± 1.29 years; BMI = 25.29 ± 3.5 kg/m2) were included. Results: We found a positive correlation between cfPWV and weight (r = 0.51; *p* < 0.0001), BMI (r = 0.44; *p* < 0.0001), WHtR (r = 0.26; *p* = 0.02), fasting insulin levels (r = 0.28; *p* = 0.02), and insulin resistance (Homeostatic Model Assessment of Insulin Resistance (HOMA-IR) index) (r = 0.25; *p* = 0.04). Multiple linear regression analysis identified BMI and HOMA-IR as independent parameters associated with cfPWV. Conclusions: Prepubescent children with obesity who were shown to be metabolically healthy presented with arterial stiffness, which is closely related to BMI and the state of insulin resistance.

## 1. Introduction

The loss of elasticity of the walls of arteries is named arteriosclerosis. Cardiovascular diseases (CVD) are a leading cause of death around the world [1] and are responsible for a significant consumption of health system resources [2]. Stiffening of the arterial wall is one major mechanism responsible for this morbidity and mortality in CVD. It begins in childhood, and cardiovascular risk factors identified at an early age tend to persist into adulthood [3].

Childhood obesity is a growing global health risk that threatens to worsen both the prevalence and morbidity-mortality of CVD in adults [4]. According to the World Health Organization, more than 340 million children and adolescents between 5 and 19 years of age were estimated to be overweight or obese in 2016 [5]. A healthy lifestyle based on the Mediterranean Diet (MedDiet) appears to be a good strategy for health, as it is associated with a lower risk of developing overweight, obesity, and abdominal obesity in the pediatric population [6]. Moreover, MedDiet has been demonstrated to yield several benefits for cardiovascular risk, showing a negative correlation with arterial stiffness (AS) in the pediatric population [7].

Measuring vascular biomarkers in order to perform cardiovascular risk stratification could be useful for identifying the pediatric population at increased cardiovascular risk. There is evidence that subclinical markers of arteriosclerosis, such as endothelial dysfunction, intima-media thickness, and AS, are abnormal in children with obesity [8]. AS reflects early vascular aging [9]. Specifically, AS is also considered a plausible predictor of arteriosclerosis’ cardiovascular morbidity and mortality. Pulse wave velocity (PWV)—a non-invasive, painless test, easy to perform in habitual clinical practice—is the gold standard for measuring AS. Indeed, higher carotid-femoral pulse wave velocity (cfPWV) values measured using this test indicate more advanced vascular aging [10]. Several studies have demonstrated the relationship between PWV values and lipid profile (total cholesterol, low-density-lipoprotein [LDL]-cholesterol, triglycerides), insulin resistance (Homeostatic Model Assessment of Insulin Resistance [HOMA-IR]), fasting insulin, and C-reactive protein (CRP) in children with obesity [11,12,13]. Therefore, the use of PWV measurements has been proposed as a useful prognostic index of vascular damage in children [14,15]. However, PWV’s role in stratifying risk in obese/overweight children has not yet been satisfactorily defined. In addition, the MHO phenotype in children has not been well studied.

With this background, our aim was to determine the relationship between AS determined by cfPWV and anthropometric, analytical, and Mediterranean Diet (MedDiet) adherence parameters in MHO children both in the total population and according to sex.

## 2. Patients and Methods

A cross-sectional study was conducted in an MHO prepubescent population aged 6–11 years. Inclusion criteria were defined according to Vukovic et al. [16]. They referred to boys (from 4 years to testicular volume <3 mL) and girls (from 4 years to Tanner S2, breast bud elevation) with obesity (age and sex-specific body mass index [BMI] ≥95th percentile) [17] and ≤1 of the following five metabolic syndrome criteria: abdominal circumference and blood pressure ≥90th percentile, triglycerides ≥150 mg/dL, high-density-lipoprotein [HDL]-cholesterol <40 mg/dL, and fasting plasma glucose ≥100 mg/dL [18]. Participants who met >1 criteria of metabolic syndrome, who were not within the age range at baseline, or who had diabetes or any metabolic disorders were excluded from the study.

Recruitment was carried out via visits to preschool and elementary school classrooms in Malaga (Andalusia, Spain). Once possible participants were selected, their parents/tutors were contacted to inform them of the study’s design and objectives as well as to request their written informed consent for their child’s voluntary participation in the study.

Anthropometric measures, questionnaires, and cfPWV were performed by trained nurses at the hospital. Weight was measured when the children were wearing light clothing, using an electronic scale (TANITA Body Composition Analyzer (Type TBF-300 MA. TANITA Corporation; 1–14–2 Maeno-cho, Itabashi-ku. Tokyo, Japan). Height was measured with no shoes, using a wall stadiometer (Stadiometer Barys Electra Model. 511-300-A0A. ASIMED). BMI (kg/m^2^) was calculated as weight (kg) divided by height (m) squared. We used the cut-off points established by the World Obesity Federation (http://www.worldobesity.org/) to define obesity as age and sex-specific BMI ≥95th percentile [19]. The waist/height ratio (WHtR) was calculated as the ratio of abdominal circumference (at the level of the mid-point between the anterosuperior iliac crest and the last costal arch, parallel to the ground and upon exhalation) and height, both in cm. Blood pressure (systolic/diastolic) was calculated as the mean of three measurements after a 5-min rest and measured using an automated electronic sphygmomanometer (OMRON M7 (HEM-780-E), OMRON Healthcare Co. Ltd., Kyoto, Japan). Mean Arterial Pressure (MAP) was calculated using the formula MAP = DBP + 1/3(SBP–DBP). MedDiet was assessed by a validated 14-item food consumption frequency questionnaire [20]. Given our prepubescent population, the last item, related to red wine intake, was omitted. Thus, high adherence was considered to be 12–13 points, moderate adherence 8–11 points, low adherence 5–7 points, and very low adherence < 5 points.

AS was measured by cfPWV using an oscillometric device (Vicorder^®^, Skidmore Medical LTD, Bristol, UK) [21]. In short, cfPWV was measured by simultaneous tonometry using an oscillometric device that allows for the recording of signals using a cuff on the lower limb and a carotid sensor (Vicorder). To perform the measurement, the subject rested for 10 min prior to the test and was placed on a stretcher in the supine position. We then proceeded to place the cuff on the lower limb at the level of the common femoral artery and the sensor at the level of the ipsilateral common carotid artery. Then, the distance of the arterial path length was calculated as the direct line distance from the sternal notch to the top of the thigh cuff, subtracting both the distance from the sternal notch to the carotid pulse and the distance measured between the thigh cuff and the femoral pulse [22]. Once the distance between both points had been obtained using, a measuring tape, the device calculated the time in milliseconds that it took for a wave to travel between the cuffs (carotid and femoral) and reported on the cfPWV by applying specific formulas that take into account systolic, diastolic, and mean blood pressures as well as the subject’s weight and height. We took three measurements and calculated the mean value for each subject. We used the cfPWV intervals and percentiles described for the normal-weight pediatric population [23].

Blood samples were taken after a 12-h fast and biochemical measurements (triglycerides, HDL-cholesterol, total cholesterol, LDL-cholesterol, glycated hemoglobin [HbA1c], insulin, and glucose) were obtained by routine methods in the Clinical Analysis Laboratory of the Regional University Hospital of Malaga. HOMA-IR was calculated as glucose (mg/dL) × insulin (μIU/mL)/405 [24,25]. We applied reference values and percentiles of fasting insulin levels (boys = 6.39 (4.69–10.09), girls = 7.8 (5.90–11.50), and HOMA-IR (boys = 1.5 (1.0–2.2), girls = 1.7 (1.3–2.6)) obtained from the European pediatric population [26].

All patients participating in the study gave their informed consent, and protocols were approved by the institutional ethics committee (Code and Approbation: 07312015. Comité de Ética de la Investigación Provincial de Malaga, belonging to the Andalusian Health Service).

### Statistical Analysis

Quantitative variables with normal distribution were expressed as mean ± standard deviation (SD). Parameters with skewed distributions such as triglycerides levels were presented as medians (25th, 75th percentile). SD (Z)-scores for anthropometric parameters (weight, height, BMI, Mean Arterial Pressure (MAP), Heart Rate (HR), and systolic and diastolic blood pressures) were calculated according to Spanish Child Growth Standards (http://www.webpediatrica.com/endocrinoped/antropometria.php).

Student’s *t*-test was used to compare quantitative variables, and the Chi-square test to compare qualitative variables. In order to determine factors that were independently associated with AS, multivariate logistic regression techniques were applied using AS as a dependent variable and controlling for confounding variables such as age, weight, BMI, SD (Z)-score for BMI, total cholesterol, LDL-cholesterol, HDL-cholesterol, fasting glucose, insulin levels, HOMA-IR index, and adherence to the MedDiet. Finally, bivariate correlations were determined using Pearson’s correlation coefficient.

Simple Interactive Statistical Analysis (SISA) was used to calculate the sample size. We assumed a 95% confidence level (0.5% error), a statistical power of 80%, and a loss rate of 5%. Based on these assumptions, a sample of 75 metabolically healthy obese prepubescent subjects was required to detect a difference in cfPWV levels between overweight/obese children of at least 1.2 m/s according to reference values in normal weight prepubescent subjects [24].

The SPSS program, version 22.0, for Windows (IBM Corporation INC. Somers, NY, USA) was used for the statistical analysis of results.

## 3. Results

During the recruitment process, 121 prepubescent children were identified, of which 46 did not meet the inclusion criteria or declined to participate. The final sample included 75 MHO children (43 boys (57.3%) and 32 girls (42.7%); *p* = 0.20) with a mean ± SD age of 10.05 ± 1.29 years (boys: 9.98 ± 1.32; girls: 10.16 ± 1.27; *p* = 0.55). As shown in Table 1C, of the 75 subjects included, 34 (45.33%) had one metabolic syndrome criterion and 41 (54.66%) did not have any.

Clinical and anthropometric characteristics (A), SD (Z)-scores (B), and analytical parameters (C) are shown in Table 1. Our population had a BMI in the range of overweight/obesity, with similar values found in both sexes (all: 25.29 ± 3.5 kg/m^2^, SDS 1.91 ± 1.12; boys: 25.48 ± 3.95 kg/m^2^, SDS 1.94 ± 1.25; girls: 25.04 ± 2.92 kg/m^2^, SDS 1.88 ± 0.92). Fasting plasma insulin levels were elevated (all: 15.58 + 7.45 µIU/mL; boys: 14.64 + 7.38 µIU/mL; girls: 16.87 + 7.48 µIU/mL), corresponding to the 97th–99th percentile in both sexes. The HOMA-IR index was also high (all: 3.08 ± 1.62; boys: 2.89 + 1.56; girls: 3.34 + 1.68), corresponding to the 95th–97th percentile in both sexes. Our population showed a moderate adherence to MedDiet (all: 9.35 ± 1.82 points; boys: 9.35 ± 1.92 points; girls: 9.34 ± 1.72 points; *p* = 0.99). Participants had increased cfPWV compared to reference values reported in the pediatric population (all: 5.90 ± 1.07 m/s; boys: 5.99 ± 1.07 m/s; girls: 5.78 ± 1.08 m/s; *p* = 0.59), corresponding to the 97th–99th percentile in both sexes.

Table 2 shows the correlation between AS measured by cfPWV and the clinical, anthropometric, and analytical parameters. Age (r: 0.23, *p* = 0.05), measurements of obesity (weight (r: 0.51, *p* < 0.0001), BMI (r: 0.44, *p* < 0.0001), WHtR (r: 0.26, *p* = 0.02)), and markers of insulin resistance (fasting insulin levels (r: 0.28, *p* = 0.02), HOMA-IR (r: 0.25, *p* = 0.04)) were positively correlated with AS in the overall population. However, differences were observed according to sex. Parameters of obesity showed a stronger positive correlation with AS in boys (weight (r: 0.59, *p* < 0.0001), BMI (r: 0.54, *p* < 0.0001), WHtR (r: 0.52, *p* = 0.001)) than in girls (weight (r: 0.37, *p* = 0.04), BMI (r: 0.26, *p* = 0.16), WHtR (r: −0.18, *p* = 0.32)). In contrast, markers of insulin-resistance were positively correlated only in girls (fasting insulin levels (r: 0.56, *p* = 0.002), HOMA-IR (r: 0.52, *p* = 0.004)), but not in boys (fasting insulin levels (r: 0.11, *p* = 0.51), HOMA-IR (r: 0.06, *p* = 0.74)). No correlation between cfPWV values and blood pressure (mean arterial pressure, heart rate, systolic and diastolic blood pressure) and MedDiet adherence were found.

After calculating the linear regression model, associations between cfPWV and anthropometric data were found. However, the analysis with clinical parameters showed that the independent determinants of AS in the overall population were SDS BMI (*β* (95% CI): 0.39 (0.15–0.63); *p* = 0.002)) and HOMA-IR (*β* (95% CI): 0.17 (0.11–0.22); *p* = 0.003). According to sex, only girls showed an association between AS and markers of insulin resistance HOMA-IR (*β* CI 95%: 0.38 (0.04–0.72), *p* = 0.03)), while the association between the parameters of obesity and cfPWV was observed only in boys: SDS BMI (β CI95%: 0.56 (0.32–0.81), *p* < 0.001)) (Table 3).

## 4. Discussion

The main finding of our work is that insulin resistance and BMI are closely related to the presence of AS in prepubescent children with obesity. Secondly, this study seriously calls into question the existence of an MHO phenotype in children. AS was found in our prepubescent population with obesity compared with children with normal weight matched by aged and sex. These findings are in concordance with previous studies [11,23,24].

The MHO phenotype is controversial and there is no consensus about what criteria should be used to define it. Although subjects with MHO present with a lower cardiometabolic risk than obese people with metabolic syndrome, they have a higher risk of CVD [27,28] and type 2 diabetes [29] compared to normal-weight people. Moreover, MHO could be considered as a transient state; indeed, over 30.6% became metabolically unhealthy obese after 10 years [30]. Finally, it is well established that MHO subjects may benefit from lifestyle modifications. Indeed, our research group has reported that weight loss after a structured lifestyle modification program based on MedDiet and exercise in premenopausal MHO women improves lipid profile [31], adipocytokines, and oxidative stress and reduces inflammatory biomarkers [32].

Currently, most authors describe MHO as an absence of the metabolic abnormalities included in the definition of metabolic syndrome [28,33] and not all definitions include the assessment of insulin resistance [34]. However, our pediatric population, despite being MHO according to the usual clinical definition, showed high levels of insulinemia and insulin resistance. This finding highlights the importance of measuring the insulin resistance status in order to diagnose MHO in children. It has been well documented that the insulin sensitivity status is correlated with a relatively low visceral fat mass, less adipose tissue dysfunction, and lower risk of dyslipidemia and hypertension [35,36,37]. In fact, the prevalence of MHO without insulin resistance is very low [35].

Children with obesity, in addition to having higher insulinemia and HOMA-IR, also present with impaired endothelial function and increased AS compared to the normal-weight population [11,12,14,38]. A good association between insulin resistance and AS measured by cfPWV has been reported in prepubescent population [12], a finding confirmed in our study, especially in girls. Differences found in the association between surrogate markers of insulin resistance and AS according to sex have been published by other authors [39].

A recent meta-analysis showed that both BMI and WHtR are useful parameters for diagnosing obesity in the pediatric population, strongly correlating with body fat measured by DEXA [40]. On the other hand, previous studies suggest that, in the general population, fat distribution may be more important than the absolute amount of fat per se in terms of increase of AS [41,42]. In this sense, sex-related differences in adipose tissue distribution may influence the association between anthropometric measures of obesity and AS assisted by cfPWV. Whereas in the male population WHtR shows the strongest association with cfPWV, in the female population, BMI is better associated with a higher cfPWV [43]. This finding is in agreement with our data, as we observed a positive association between AS and parameters of obesity (BMI, WHtR) in boys.

We did not find any relationship between adherence to MedDiet and AS in our prepubescent MHO population. This finding is in contrast to previous studies, which show a negative correlation between AS and a healthy lifestyle based on MedDiet and physical activity in both adults [44,45,46] and children [4]. Furthermore, a lifestyle modification program in obese youth may improve AS along with reducing HOMA-IR [47].

AS is a reliable marker of vascular damage and an independent predictor of CVD [48]. Therefore, increased AS in children with obesity not only seriously challenges the existence of the MHO phenotype in children, but also suggests that obese children may have early subclinical vascular damage and a high risk of further CVD. Accordingly, the prompt implementation of preventive measures such as promoting a healthy and active lifestyle in children with obesity emerges as a critical priority for public health policies.

### Limitations

First, our study was performed on a small Caucasian population sample, so it cannot be extrapolated to other ethnic groups. Second, there was no randomization or control group (normal-weight subjects). Third, we did not study other variables that could potentially influence the insulin-resistance status, such as physical activity. Fourth, it would have been necessary to study whether the differences in adipose tissue distribution according to sex influences results related to AS and anthropometric parameter changes. Finally, it has been described that cfAS changes during childhood [49]; our results showed that cf-PWV tends to increase in early adolescence. In order to avoid this potential bias, we included prepubescent children and used reference values recommended for cfPWV in the pediatric population [23]. Our data demonstrated that, in our population, the cardiovascular risk of prepubescent children <9 years old with high cfPWV values has been underestimated. However, there have been few studies performed in the pediatric population with an MHO phenotype.

## 5. Conclusions

In conclusion, arterial stiffness was closely related to BMI and insulin resistance status in a population of MHO prepubescent children in our environment. cfPWV could be useful as a non-invasive tool to identify the presence of arterial stiffness in the prepubescent population with obesity.

## Figures and Tables

**Table 1 ijerph-17-06995-t001:** Clinical parameters (**A**), SD (Z)-scores (**B**), and analytical (**C**) parameters and metabolic syndrome criteria, total and by sex (Mean ± SD).

**A**	**All (*n* = 75)**	**Boys (*n* = 43; 57.3%)**	**Girls (*n* = 32; 42.7%)**	***p***
Age (years)	10.05 ± 1.29	9.98 ± 1.32	10.16 ± 1.27	0.49
Weight (kg)	57.13 ± 12.15	58.07 ± 12.69	55.86 ± 11.44	0.70
BMI (kg/m^2^)	25.29 ± 3.5	25.48 ± 3.95	25.04 ± 2.92	0.38
WHtR	0.56 ± 0.06	0.57 ± 0.07	0.55 ± 0.05	0.27
Growth Percentile	99.55 ± 5.31	94.65 ± 6.22	96.75 ± 3.51	0.09
SBP (mmHg)	106 ± 13	106 ± 14	107 ± 12	0.27
DBP (mmHg)	70 ± 12	70 ± 12	69 ± 13	0.89
MAP (mmHg)	81.92 ± 11.75	81.79 ± 11.81	82.09 ± 11.86	0.91
HR (bpm)	81.99 ± 12.50	79.36 ± 11.51	85.44 ± 13.08	0.04
Adherence to MedDiet (points)	9.35 ± 1.82	9.35 ± 1.92	9.34 ± 1.72	0.99
cfPWV (m/s)	5.70 (5.30–6.20)	5.8 (5.4–6.4)	5.6 (5.2–6.0)	0.59
**B**	**All (*n* = 75)**	**Boys (*n* = 43; 57.3%)**	**Girls (*n* = 32; 42.7%)**	***p***
Height	0.87 ± 1.08	0.98 ± 0.90	0.72 ± 1.28	0.1
Weight	1.97 ± 1.30	2.00 ± 1.33	1.92 ± 1.27	0.7
BMI	1.91 ± 1.12	1.94 ± 1.25	1.88 ± 0.92	0.1
SBP	0.09 ± 1.22	−0.02 ± 1.31	0.25 ± 1.11	0.3
DBP	0.63 ± 1.09	0.60 ± 1.02	0.67 ±1.20	0.6
**C**	**All (*n* = 75)**	**Boys (*n* = 43; 57.3%)**	**Girls (*n* = 32; 42.7%)**	***p***
Glucose (mg/dL) NV: 70–110 mg/dL	79.32 ± 9.00	79.59 ± 7.05	78.97 ± 11.19	0.07
HbA1c (%) NV: 4.0–6.0 %	5.25 ± 0.27	5.23 ± 0.26	5.28 ± 0.28	0.61
Fasting insulin (µIU/mL) NV: 4–16 µUI/mL	15.58 ± 7.45	14.64 ± 7.38	16.87 ± 7.48	0.65
HOMA-IR NV: ≤3.4	3.08 ± 1.62	2.89 ± 1.56	3.34 ± 1.68	0.36
Total Cholesterol (mg/dL) NV: <200 mg/dL	151.07 ± 26.87	154.12 ± 30.37	147.03 ± 21.12	0.03
LDL-cholesterol (mg/dL) NV: <130 mg/dL	85.00 ± 22.19	88.85 ± 25.45	79.90 ± 15.97	0.03
HDL-Cholesterol (mg/dL) NV: >50 mg/dL	50.03 ± 10.74	49.76 ± 10.87	50.39 ± 10.75	0.76
Triglycerides (mg/dL) NV: <150 mg/dL	67.50 (53.00–96.75)	67.00 (48.00–89.50)	68.00 (58.00–104.00)	0.57
Metabolic Syndrome Criteria				
Abdominal Circumference (≥90th percentile)	8 (10.7%)	6 (14.0%)	2 (6.3%)	
SBP (>130 mg/dL)	0 (0.0%)	0 (0.0%)	0 (0.0%)	
DBP (>85 mg/dL)	8 (10.7%)	6 (14.0%)	2 (6.3%)	
Glucose (≥100 mg/dL)	3 (4.1%)	1 (2.3%)	2 (6.5%)	
HDL-Cholesterol (<40 mg/dL)	10 (13.5%)	6 (14%)	4 (12.9%)	
Triglycerides (>150 mg/dL)	5 (6.8%)	3 (7%)	2 (6.5%)	

NV: Normal Values, BMI: Body Mass Index, WHtR: Waist/Height Ratio, SBP: Systolic Blood Pressure; DBP: Diastolic Blood Pressure, MAP: Mean Arterial Pressure. HR: Heart Rate. HbA1c: Hemoglobin A1c. HOMA-IR: Homeostasis Model Assessment of Insulin Resistance. MedDiet: Mediterranean diet, cfPWV: carotid-femoral Pulse Wave Velocity (reference levels of cfPWV normal-weight: boys 5.31 ± 0.04 m/s, girls 5.14 ± 0.07 m/s) [23].

**Table 2 ijerph-17-06995-t002:** Pearson correlation between carotid-femoral pulse wave velocity and clinical, anthropometric, and analytical parameters.

Parameters	All		Boys		Girls	
	r	*p*	r	*p*	r	*p*
Weight (kg)	0.50	<0.001	0.59	<0.001	0.18	0.04
SDS BMI	0.23	<0.001	0.40	0.007	0.18	0.33
WHtR	0.25	0.03	0.50	<0.001	−0.18	0.32
MAP (mmHg)	−0.01	0.94	−0.22	0.16	0.26	0.14
HR (bpm)	−0.03	0.79	0.04	0.80	−0.06	0.75
Glucose (mg/dL)	−0.03	0.81	−0.18	0.27	0.09	0.65
HbA1c (%)	−0.07	0.57	0.12	0.44	−0.29	0.12
HOMA-IR	0.25	0.04	0.06	0.74	0.52	0.004
Total cholesterol (mg/dL)	−0.07	0.56	−0.04	0.83	−0.18	0.33
HDL-cholesterol (mg/dL)	−0.008	0.94	−0.05	0.78	0.05	0.80
LDL-cholesterol (mg/dL)	−0.05	0.66	−0.02	0.92	−0.20	0.27
Triglycerides (mg/dL)	−0.09	0.45	−0.02	0.88	−0.17	0.35

SDS-BMI: SDS-Body Mass Index, WHtR: Waist/Height Ratio. SBP: MAP: Mean Arterial Pressure. HR: Heart Rate. HOMA-IR: Homeostasis Model Assessment of Insulin Resistance.

**Table 3 ijerph-17-06995-t003:** Determinants of carotid-femoral pulse wave velocity. Multiple linear regression model.

Parameters	All		Boys		Girls	
	*β*	*p*	*β*	*p*	*β*	*p*
Age (years)	0.29 (0.08–0.49)	0.06	0.19 (−0.05–0.44)	0.12	0.18 (−0.12–0.49)	0.23
SDS BMI	0.39 (0.15–0.63)	0.002	0.56 (0.32–0.81)	<0.001	0.17 (−0.06–0.09)	0.66
MAP (mmHg)	−0.01 (−0.03–0.01)	0.26	−0.03 (−0.06–0.01)	0.08	0.015 (−0.21–0.05)	0.40
HR (bpm)	−0.001 (−0.02–0.2)	0.96	0.01 (−0.02–0.02)	0.95	−0.01 (−0.05–0.02)	0.38
HOMA-IR	0.17 (0.11–0.22)	0.003	−0.07 (−0.26–0.12)	0.48	0.38 (0.04–0.72)	0.03

The values shown are adjusted linear regression coefficients (*β*) and 95% confidence intervals, estimated with pulse wave velocity as a dependent variable. SDS-BMI: SD-Body Mass Index, MAP: Mean Arterial Pressure. HR: Heart Rate. HOMA-IR: Homeostasis Model Assessment of Insulin Resistance.

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
