# Peer review of "Metabolically Healthy Obesity: Presence of Arterial Stiffness in the Prepubescent Population"

_ijerph, 2020, doi:10.3390/ijerph17196995_

Round 1

Reviewer 1 Report

In the present form the manuscript is acceptable for publications. The authors included each of the commemts and sugestins indicated by the reviewers.

Author Response

Thank you very much for you revision

Reviewer 2 Report

Many thanks for the opportunity to re-review this paper.

The paper reads much better, but unfortunately I still don't feel that the analyses conducted are entirely appropriate for the question being addressed, and have documented some of these concerns below.

It is still not clear how you defined MHO. Methods state abdominal circumference and BP >90th whereas Table 1 shows WC >86m (actually says < but assuming typo) and BP over 130/85).

Your statistical analysis of the data should go from a more basic assessment to a more detailed one. Tables 2 and 3 need to be swapped as Table 3 is looking at simple bivariate correlations and Table 2 is expanding upon this to try to account for potential confounding using a more detailed methodology. The presence of a significant correlation in table 3 is relatively meaningless if it has already been displayed in table 2 to likely arise from confounding by other predictors.

With regards to Table 2, as previously mentioned by myself and I think also another reviewer, the choice of variables for the multiple linear regression analysis is inappropriate. There are a number of reasons for this. Firstly, a general ‘rule of thumb’ for regression analysis is that there should be no more than one predictor variable for every 10 or so participants to avoid overfitting the model. Your model has 18 variables for a cohort of 75 people, which is then split roughly in half for males/females, so you have far too many predictors to be reliable. Secondly, many of the predictors included are correlated and so it doesn’t make sense to include them together. In the simplest terms, your model is trying to estimate the independent effect of one predictor if all other included variables are held constant. You’re therefore asking in Table 2 what effect the BMI SDS of two individuals has on PWV if they are both the same weight and the same BMI measured in kg/m2 – it doesn’t really make sense. Same goes for looking at effect of MAP when SBP and DBP (which make up MAP) are the same. Ideally you need to propose a model with a limited number of exposures which you expect may influence the outcome. This for example could include age, sex, BMI SDS, MAP, HR, and HOMA-IR – all risk factors which have a physiological basis for being important for PWV. Perhaps some advice from a third party statistician would help to produce a more valid regression model.   

Apologies if I’ve asked this before, but is the table referring to metabolic syndrome criteria showing how many of your population had one metabolic abnormality – e.g. 3 people had high glucose but were fine for everything else, 5 had high triglycerides alone, etc? If so, may be good to state somewhere in text that 34 participants had one component of MetS and 41 were free from all.

Line 259 – Can’t state an exact age when AS begins to increase (e.g. 10.4 in girls and 12.1 in boys). Best say something along lines of ‘tends to increase in early adolescence’

Author Response

Journal:IJERPH (ISSN 1660-4601)

Manuscript ID:ijerph-879461

Type:Article

Number of Pages:15

Title:Metabolically healthy obesity: Presence of arterial stiffness in the prepubescent population

Authors:M Isabel Ruiz-Moreno, Alberto Vilches-Perez , Cristina Gallardo-Escribano , Antonio Vargas-Candela , M Dolores Lopez-Carmona , Luis Miguel Pérez-Belmonte , Alejandro Ruiz-Moreno , Ricardo Gomez-Huelgas, M Rosa Bernal-Lopez *

Abstract

Aim: Atherosclerotic cardiovascular disease, one of the world’s leading causes of death, first manifests itself at an early age. The identification of children who may have increased cardiovascular risk in the future could be an important prevention strategy. Our aim was to assess the clinical, analytical, and dietary variables associated with arterial stiffness (AS), measured by pulse wave velocity (PWV) in a prepubescent population with metabolically healthy obesity (MHO). Subjects and Methods: A cross-sectional study in prepubescent subjects with obesity who had £1 metabolic syndrome criteria (abdominal perimeter and blood pressure ≥90th percentile, triglycerides >150 mg/dL, HDL-cholesterol <40 mg/dL, fasting plasma glucose ³100 mg/dL) were conducted. Adherence to Mediterranean Diet, blood pressure, BMI, waist/height ratio (WHtR), glycemic status, lipid profile, and carotid-femoral PWV were analyzed. 75 MHO children (boys: 43; girls: 32; p=0.20) (age=10.05±1.29 years; BMI=25.29±3.5 kg/m2) were included. Results: We found a positive correlation between carotid-femoral PWV and weight (r=0.51; p<0.0001), BMI (r=0.44; p<0.0001), WHtR (r=0.26; p=0.02), fasting insulin levels (r=0.28; p=0.02), and insulin resistance (HOMA-IR index) (r=0.25; p=0.04). Multiple linear regression analysis identified BMI and HOMA-IR as independent parameters associated with PWV. Conclusions: In MHO prepubescent children, BMI and insulin-resistance status are related to arterial stiffness. PWV could potentially be a useful non-invasive technique to identify cardiovascular risk in childhood.

Final del formulario

Principio del formulario

Review Report Form

Open Review

(x) I would not like to sign my review report

( ) I would like to sign my review report  English language and style

() Extensive editing of English language and style required 

() Moderate English changes required 

(x) English language and style are fine/minor spell check required 

( ) I don't feel qualified to judge about the English language and style 

Yes

Can be improved

Must be improved

Not applicable

Does the introduction provide sufficient background and include all relevant references?

(x)

( )

( )

( )

Is the research design appropriate?

( )

( )

(x)

( )

Are the methods adequately described?

( )

(x)

( )

( )

Are the results clearly presented?

( )

( )

(x)

( )

Are the conclusions supported by the results?

( )

(x)

( )

( )

Comments and Suggestions for Authors

Many thanks for the opportunity to re-review this paper.

The paper reads much better, but unfortunately I still don't feel that the analyses conducted are entirely appropriate for the question being addressed, and have documented some of these concerns below.

It is still not clear how you defined MHO. Methods state abdominal circumference and BP >90thwhereas Table 1 shows WC >86m (actually says < but assuming typo) and BP over 130/85).

Response: The reviewer is right. It was a typo mistake that has been modified in Table 1.

There exist different criteria to define Metabolically Healthy Obese population. As it is described in the text (Patient and Methods section), the authors considered metabolically healthy obese those subjects with < 1 criteria of metabolic syndrome: abdominal circumference and blood pressure ≥90th percentile, triglycerides ≥150 mg/dL, high-density-lipoprotein [HDL]-cholesterol <40 mg/dL, and fasting plasma glucose ≥100 mg/dL; according to the following reference: Vukovic R, Dos Santos TJ, Ybarra M, Atar M. Children With Metabolically Healthy Obesity: A Review. Front Endocrinol (Lausanne). 2019;10:865. Published 2019 Dec 10. doi:10.3389/fendo.2019.00865. The reviewer could find a consensus-based definition of MHO in children in Table 1 on this reference.

The authors modified this sentence in “Patient and Methods” section as: “A cross-sectional study was conducted in a MHO prepubescent population aged 6–11 years. Inclusion criteria were defined according to Vukovic et al[16].  These were, boys (from 4 years to testicular volume <3 ml) and girls (from 4 years to Tanner S2, breast bud elevation) with obesity (age- and sex-specific body mass index [BMI] ≥95th percentile)[17] and £1 of the following 5 metabolic syndrome criteria: abdominal circumference and blood pressure ≥90th percentile, triglycerides ≥150 mg/dL, high-density-lipoprotein [HDL]-cholesterol <40 mg/dL, and fasting plasma glucose ³100 mg/dL[18].”

Your statistical analysis of the data should go from a more basic assessment to a more detailed one. Tables 2 and 3 need to be swapped as Table 3 is looking at simple bivariate correlations and Table 2 is expanding upon this to try to account for potential confounding using a more detailed methodology. The presence of a significant correlation in table 3 is relatively meaningless if it has already been displayed in table 2 to likely arise from confounding by other predictors.

With regards to Table 2, as previously mentioned by myself and I think also another reviewer, the choice of variables for the multiple linear regression analysis is inappropriate. There are a number of reasons for this. Firstly, a general ‘rule of thumb’ for regression analysis is that there should be no more than one predictor variable for every 10 or so participants to avoid overfitting the model. Your model has 18 variables for a cohort of 75 people, which is then split roughly in half for males/females, so you have far too many predictors to be reliable. Secondly, many of the predictors included are correlated and so it doesn’t make sense to include them together. In the simplest terms, your model is trying to estimate the independent effect of one predictor if all other included variables are held constant. You’re therefore asking in Table 2 what effect the BMI SDS of two individuals has on PWV if they are both the same weight and the same BMI measured in kg/m2 – it doesn’t really make sense. Same goes for looking at effect of MAP when SBP and DBP (which make up MAP) are the same. Ideally you need to propose a model with a limited number of exposures which you expect may influence the outcome. This for example could include age, sex, BMI SDS, MAP, HR, and HOMA-IR – all risk factors which have a physiological basis for being important for PWV. Perhaps some advice from a third party statistician would help to produce a more valid regression model.   

Response: The authors have modified the order of tables 2 and 3. In addition, variables included in the statistical regression model have been modified. Authors have included the variables as suggest the reviewer.

The rest of variables in previous revisions were included because other reviewer solicited it. Now, the authors deleted the no significant variables.

The “Results” section has been modified as: “After calculating the linear regression model, associations between cfPWV and anthropometric data were found.  However, the analysis with clinical parameters showed that the independent determinants of AS in the overall population were SDS BMI [β (95%CI): 0.39 (0.15-0.63); p=0.002)] and HOMA-IR [β (95%CI): 0.17 (0.11-0.22); p=0.003]. According to sex, only girls showed an association between AS and markers of insulin resistance HOMA-IR [β CI95%: 0.38 (0.04-0.72), p=0.03]), while the association between the parameters of obesity and cfPWV was observed only in boys: SDS BMI [β CI95%: 0.56 (0.32-0.81), p<0.001]) (Table 3).

The new table reflect the results of the multiple regression model has been modified as:

Table 3. Determinants of carotid-femoral pulse wave velocity. Multiple linear regression model.

Parameters

All

Boys

Girls

β

p

β

p

β

p

Age (years)

0.29 (0.08-0.49)

0.06

0.19 (-0.05-0.44)

0.12

0.18 (-0.12-0.49)

0.23

SDS BMI

0.39 (0.15-0.63)

0.002

0.56 (0.32-0.81)

<0.001

0.17 (-0.06-0.09)

0.66

MAP (mmHg)

-0.01 (-0.03-0.01)

0.26

-0.03 (-0.06-0.01)

0.08

0.015 (-0.21-0.05)

0.40

HR (bpm)

-0.001 (-0.02-0.2)

0.96

0.01 (-0.02-0.02)

0.95

-0.01 (-0.05-0.02)

0.38

HOMA-IR

0.17 (0.11-0.22)

0.003

-0.07 (-0.26-0.12)

0.48

0.38 (0.04-0.72)

0.03

The values shown are adjusted linear regression coefficients (β) and 95% confidence intervals, estimated with pulse wave velocity as a dependent variable. SDS-BMI: SD-Body Mass Index, MAP: Mean Arterial Pressure. HR: Heart Rate. HOMA-IR: Homeostasis Model Assessment of Insulin Resistance.

Apologies if I’ve asked this before, but is the table referring to metabolic syndrome criteria showing how many of your population had one metabolic abnormality – e.g. 3 people had high glucose but were fine for everything else, 5 had high triglycerides alone, etc? If so, may be good to state somewhere in text that 34 participants had one component of MetS and 41 were free from all.

Response: Considering the recommendation of the reviewer, the authors have included the following information in “Results” section clarifying the data shown in table 1C as: “As it is shown in Table 1C, of the 75 subjects included, 34 (45,33%) had one metabolic syndrome criterion and 41 (54,66%) did not have any.

Line 259 – Can’t state an exact age when AS begins to increase (e.g. 10.4 in girls and 12.1 in boys). Best say something along lines of ‘tends to increase in early adolescence’

Response: Considering the recommendation from the reviewer, the authors have added the following information in “Limitation” section: “Finally, it has been described that cfAS changes during childhood [51]; our results showed that cf-PWV tends to increase in early adolescence.

Submission Date: 07 September 2020

Date of this review: 10 Sep 2020 18:16:30

Reviewer 3 Report

All comments were appropriately addressed.

Author Response

Thank you for your revision

Round 2

Reviewer 2 Report

Many thanks once again for addressing my comments. Nothing further to add. 

This manuscript is a resubmission of an earlier submission. The following is a list of the peer review reports and author responses from that submission.

Round 1

Reviewer 1 Report

The authors evaluated anthropometric, analytical, and dietary variables associated  with arterial stiffness for to identificate of children who may have an increased in ACVD. The current study examine the relationsips between anthropometric characteristics and analytical parameters with cfPWV value, and demonstrated that insulin resistance and BMI are closely related to the presence of arterial stiffness in prepubescent MHO children.

While this study is well done and could be of value to this field of research. It is necessary to include a group of children of the same ages and sex with BMI values of normoweight in order to validate determination of the cfPWV as a good marker of risk for ACVD.

Author Response

Journal: IJERPH (ISSN 1660-4601)

Manuscript ID: ijerph-879461

Type: Article

Number of Pages: 15

Title: Metabolically healthy obesity: Presence of arterial stiffness in the prepubescent population

Authors: M Isabel Ruiz-Moreno, Alberto Vilches-Perez , Cristina Gallardo-Escribano , Antonio Vargas-Candela , M Dolores Lopez-Carmona , Luis Miguel Pérez-Belmonte , Alejandro Ruiz-Moreno , Ricardo Gomez-Huelgas, M Rosa Bernal-Lopez *

We thank you very much for giving us the opportunity to revise our manuscript again. We have carefully considered the comments made by the reviewer and agree with most of them. Each comment has been addressed and we have modified the manuscript accordingly. We sincerely hope that the current version of the manuscript will be acceptable for publication in your journal. All changes are shown in red so that they may be easily seen.

Dr. María Rosa Bernal-Lopez

Reviewer 2 Report

This paper by Ruiz-Moreno et al sets out to identify children who may have increased risk of future CVD by assessing clinical, analytical, and dietary variables associated with PWV. The authors conclude that in pre-pubescent children with metabolically healthy obesity, BMI and IR are related to PWV, and that PWV may be a good technique for assessing risk at this age. In general the paper is nicely-written, but I feel that the overall message could be clarified a bit better to justify why the data are important. The authors conclusion is that PWV could be a good technique to identify CV risk in children, but their data don’t really show this (for this statement to be true, they would have to show that those with higher PWV went on to have worse CV outcomes). What they show instead is that lots of young children who are obese appear to have very few metabolic abnormalities at this age and so therefore may be regarded as ‘metabolically healthy’. However, there is actually already evidence of what we know are adverse arterial phenotypes even in these people, so perhaps this condition isn’t as benign as some may say. To me it would make much more sense to phrase the paper in this way and write intro and discussion accordingly, and to focus on the fact that the evidence suggests that if you manage their weight as they transition into adulthood, it seems you can probably reverse this before it’s too late. This would depend on the findings holding up when  a number of concerns I have about some of the analyses carried out are addressed, however, and I have listed these below along with some other points.

Major Points:

  • Appreciate this may not be possible, but the paper would be greatly strengthened by having some normal-weight individuals as controls to compare the MHO people to. While PWV is relatively simple to carry out in principle, methodological differences in measuring the distance between sites (what part of cuff do you measure to – top/middle/bottom? How much effort is made to account for increased distance in obesity caused by large stomach etc) means that sometimes it can be difficult to compare to reference values. Looking at the values here, even though we are looking at obese people, they do look pretty big for this age (I’d guess all well above 99th percentile shown in Thurn et al reference charts) and an in-house comparison would provide much stronger data.

  • Related to the above point, having normal weight individuals would also give a better indication of the importance of BMI on PWV. You show they are related here, but it is tricky to interpret how relevant this is given that you have already restricted your popualtion to the top 5% of obese individuals (so the range of BMI here is basically going only from very obese to very very obese)

  • One of the major limitations to the analyses here is the failure to account for blood pressure (particularly MAP) when assessing relationships between clinical characteristics and PWV. It is well-established at a young age that much (if not all) of the increase in PWV is due to a ‘functional’ stiffening of the arteries (caused by higher blood pressure distending the vessel) rather than actual ‘structural’ adaptations causing stiffer walls. In fact, some studies have reported that when MAP is accounted for, PWV may even be improved in obese participants due to the chronic hyperaemic state which comes with having more body mass. The authors have not investigated associations between MAP and PWV in their correlations, and have not accounted for it in regressions, so the risk of confounding here is high.

  • Another issue with the regression analysis is that age and sex are not accounted for. These are children at varying stages of development where interpretation of BMI and IR depend heavily on age, so these must be included as well.

  • The criteria for stratifying and analysing obesity is unclear. BMI at the ages studied here varies depending on both age and sex, and it is good to see that the authors have stated in their methods that they considered those above 95th percentile of age- and sex-adjusted values to be obese. However, the values then reported in the clinical characteristics table appear to be basic kg/m2 values as used in adults, and so the claim that both boys and girls were either overweight or obese probably doesn’t make much sense here as these values cannot be applied to children in the same way (your children are apparently all over 95th percentile for obesity – they’re huge). Same goes for the regression analyses, the use of normal BMI values here is likely inappropriate if this is what was used? Perhaps a better approach would be to calculate BMI age- and sex-specific z-scores using the LMS method and use these values for all analyses referring to BMI.

  • The reference values used to compare PWV are using a different device and are therefore likely not directly comparable. A large reference dataset using the VIcorder device does exist and should be used instead (Thurn et al 2015 American Journal of Hypertensiom 28(12): 1480).

Minor Points:

  • It’s not entirely clear how you attempted to recruit participants. Did you specifically target obese kids? It says 46 out of 121 were not eligible, is this because they weren’t obese? Or had too many adverse MetS components? Or both?
  • Methods state that abdominal obesity and BP above 90th percentile classed as metabolically unhealthy, but tables show waist circumference 86cms and BP > 130 and 85mmHg. Which is it?
  • Table 2 doesn’t include any anthropometric data
  • Pg 3 – “information about each participants personal and family history was obtained”. Was this used for anything?

Author Response

(The authors gave the same response as above.)

Author Response

(The authors gave the same response as above.)

Round 2

Reviewer 1 Report

The work is acceptable in the present format.

Reviewer 2 Report

Many thanks for taking the time to address my original comments. However, I still have a number of concerns which I feel continue to preclude the publication of the manuscript in its current form.

There appear to be two major conclusions which the authors have drawn from this paper, both of which are probably true given what is known already in the literature, but neither of which I’m sure are really adequately supported by the data here.

Firstly, the authors main conclusion is that arterial stiffness is present in metabolically healthy O/O children compared to their normalweight peers. However, this is based solely on comparison to a reference dataset which, as I’ve already stated in my original comments, I feel is problematic based upon the difficulty in comparing different studies with different devices, measurement techniques, operators, etc. Regardless, they don’t appear to have stated in the methods in their paper what dataset they compared their participants to, and I’m confused by their response in the reviewer feedback as they list three separate studies (22, 23, 27).

Secondly, they also conclude that BMI and HOMA-IR are independently related to PWV, but I remain puzzled by the various analyses they have presented in their reviewer responses and in the paper itself to justify this. I have the following points to make about this particular issue below:

  • In order to investigate factors potentially related to PWV, the best approach to a statistical model would likely be to select a number of exposures of interest chosen a priori based on the expectation that they will be influential, and then test these in a multivariable model to see if they are. The authors talk repeatedly in this manuscript about the potential influence of factors such as Mediterranean diet, but never include them in any analyses.
  • The authors report that BMI is related to PWV, but they appear to be using normal kg/m2 BMI as the exposure here, which isn’t accurate in children. They have now calculated BMI z-scores as requested by a number of reviewers, but appear to have entered this into a multivariable model in Table 2 alongside the other BMI, and then reported the results of the adult BMI anyway, which is incorrect.
  • Why don’t girls have BMI measured in Table 2, and why don’t boys have insulin?
  • The authors still haven’t accounted for MAP in a fully adjusted regression model to assess its impact. In the reviewer responses they have simultaneously adjusted for SBP and DBP in a model with only a limited number of anthropometric values.
  • Why have the authors followed their more detailed regression analysis in Table 2 (which is really the model which will most accurately answer their research question) with a series of basic correlations in Table 3, and what is the rationale for choosing these relationships which don’t include age, BMI, etc?

Other comments:

Lot of citations appear to refer to wrong references (e.g. citatation number 27 in text appears to actually refer to citation 26 in reference list)

Reviewer 3 Report

The authors made numerous amendments to improve the manuscript. However, some critical points still needs attention. 

  1. The authors respectfully highlighted that there is scientific evidence to support that Arterial Stiffness as a marker of atherosclerosis and provided 3 citations for this statement. Yet, in the paper of Medda et al. (2020) they did not test whether arterial stiffness (aortic PWV) is a measure of atherosclerosis. They explicitly wrote: "This study explores the association between personality, carotid atherosclerosis and arterial stiffness, and the contribution of genes and environment to this association." Their focus was on each of these measures in separation with a specific aim to investigate associations with psychological factors. The second reference to the paper of Stoner et al. (2020) was also incorrectly used to defend this statement. The authors clearly wrote in their first sentence of the paper: "Carotid-femoral pulse wave velocity (cfPWV) is widely used in epidemiological studies to assess central arterial stiffness." The terminology of atherosclerosis and arteriosclerosis were used used interchangeably for years and incorrectly so. The true definitions of these conditions are clearly explained in several state-of-the-art reviews. There is of course no doubt that a strong association exists between atherosclerosis and arteriosclerosis, but these are two separate pathologies, and can co-exist. Mostly, atherosclerosis does not exist without associated arterial stiffness - this is where the confusion enters the argument.
  2. Regarding the BP device model used in an adolescent study, there are concerns to be address. There is no evidence currently that indicates pediatric validation for the HEM-780-E. This OMRON device is only validated in adults (by ESH-IP and BHS protocol in 2007 and 2008) and in preeclampsia/pregance (by BHS protocol in 2009). The authors need to justify the use of a BP device not fit for pediatric BP evaluation.
  3. Regarding the comment on the adjustment of mean arterial pressure (MAP), the authors may have misunderstood. The adjustment for mean arterial pressure (MAP) is always recommended in multiple regression (see paper: Mikael Gottsa ̈ ter, Gerd O ̈ stling, Margaretha Persson, Gunnar Engstro ̈ m, Olle Melander, and Peter M. Nilsson. Non-hemodynamic predictors of arterial stiffness after 17 years of follow-up: the Malmo« Diet and Cancer study. Journal of Hypertension 2015, 33:957–965). In addition, I think that adjustment for heart rate (HR) can also be recommended (= both MAP and HR). The reason for this is that it reflects sympathetic nervous activation and this should be avoided by very calm and relaxed conditions in the first place when the PWV is measured. However, to further adjust for HR will even further standardise the evaluation of PWV, but not all experts agree on this. Therefore, my suggestion is to review differences in PWV associations while adjusting for MAP and not SBP or DBP.
  4. The authors perhaps also misunderstood the comment about "Parameters of obesity showed a stronger positive correlation with carotid-femoral arterial stiffness in boys…”, yet, z-transformations were not performed to determine the strength of the correlation coefficients. The response of the authors were that z-scores of the anthropometrics were included, but this is not what the comment was suggesting. In order to say one correlation is stronger in group X than group Y, one needs to perform transformation, also known as Fisher’s r to z transformation. This determines if those correlation coefficients are truly "stronger" in one compared to another group. E.g. the beta coefficient for weight  in boys was 0.05 and for girls 0.04. Are these two coefficients statistically different from each other? The same statement is on page 5: "the association between parameters of obesity and cfPWV was stronger in boys BMI: [β CI95%: 0.17 (0.09-0.25)], p<0.0001]) (Table 2)."
  5. The authors should comment on including both SBP and DBP in the same models, also why include the SDS if its corresponding variable is already included in the model? This is causing collinearity due to the strong relationship between SBP and DBP. Rather include MAP only.